# The Evolution of Ki-67 and Breast Carcinoma: Past Observations, Present Directions, and Future Considerations

**DOI:** 10.3390/cancers15030808

**Published:** 2023-01-28

**Authors:** Brian S. Finkelman, Huina Zhang, David G. Hicks, Bradley M. Turner

**Affiliations:** Department of Pathology and Laboratory Medicine, University of Rochester Medical Center, 601 Elmwood Ave., Rochester, NY 14620, USA

**Keywords:** Ki-67, breast cancer, proliferation, prognostic, predictive, analytic validity, clinical validity, standardization, monarchE, abemaciclib

## Abstract

**Simple Summary:**

Cellular proliferation is a central determinant of breast cancer recurrence risk and response to chemotherapy. The discovery of the Ki-67 antibody by Dr. Johannes Gerdes in 1983, with the observation that Ki-67 immunohistochemical expression was only present in proliferating cells, opened the door for considerations about using Ki-67 immunohistochemical expression for the evaluation of breast cancer recurrence risk, as well as for the evaluation of how well patients with breast cancer might respond to chemotherapy. Unfortunately, the lack of consistently reproducible Ki-67 results between pathologists has limited the use of Ki-67 for these prognostic and predictive evaluations in routine clinical practice. We review how the use of Ki-67 in breast cancer has evolved over the past 40 years, with a summary of the present literature on Ki-67 in breast cancer, and a discussion on the future of Ki-67 as a prognostic and predictive marker for breast cancer in clinical practice.

**Abstract:**

The 1983 discovery of a mouse monoclonal antibody—the Ki-67 antibody—that recognized a nuclear antigen present only in proliferating cells represented a seminal discovery for the pathologic assessment of cellular proliferation in breast cancer and other solid tumors. Cellular proliferation is a central determinant of prognosis and response to cytotoxic chemotherapy in patients with breast cancer, and since the discovery of the Ki-67 antibody, Ki-67 has evolved as an important biomarker with both prognostic and predictive potential in breast cancer. Although there is universal recognition among the international guideline recommendations of the value of Ki-67 in breast cancer, recommendations for the actual use of Ki-67 assays in the prognostic and predictive evaluation of breast cancer remain mixed, primarily due to the lack of assay standardization and inconsistent inter-observer and inter-laboratory reproducibility. The treatment of high-risk ER-positive/human epidermal growth factor receptor-2 (HER2) negative breast cancer with the recently FDA-approved drug abemaciclib relies on a quantitative assessment of Ki-67 expression in the treatment decision algorithm. This further reinforces the urgent need for standardization of Ki-67 antibody selection and staining interpretation, which will hopefully lead to multidisciplinary consensus on the use of Ki-67 as a prognostic and predictive marker in breast cancer. The goals of this review are to highlight the historical evolution of Ki-67 in breast cancer, summarize the present literature on Ki-67 in breast cancer, and discuss the evolving literature on the use of Ki-67 as a companion diagnostic biomarker in breast cancer, with consideration for the necessary changes required across pathology practices to help increase the reliability and widespread adoption of Ki-67 as a prognostic and predictive marker for breast cancer in clinical practice.

## 1. Introduction

Proliferation is a central determinant of prognosis and response to chemotherapy in patients with breast cancer [1]. Breast cancers with high levels of proliferation can be expected to have a more aggressive clinical course, but also to show an improved response to adjuvant (postsurgical) or neoadjuvant (presurgical) chemotherapy. The evaluation of cellular proliferation in breast cancer cells has been measured using several different methodological approaches, including analysis of thymidine uptake [2,3], flow cytometry to determine the percent of cells in S phase [2,4], the counting of mitotic figures, the immunohistochemical expression of Ki-67 [4,5], and through molecular multigene assays which measure genes transcripts associated with proliferation as a central component of their genomic profile (such as Oncotype DX and MammaPrint) [6,7,8].

For almost 40 years now, the monoclonal antibody Ki-67 has been proposed to be a valuable biomarker for assessing cellular proliferation in breast cancer and other solid tumor tissue samples across many pathology laboratories [5,9]. This non-histone nuclear antigen is significantly expressed during the G_1_, S, G_2_, and mitotic phase of the cell cycle (through standard immunohistochemistry methodologies) [9], with only very low and relatively insignificant expression levels in the quiescent (G_0_) and the early G_1_ phase, not detectable by standard immunohistochemistry methodology [10]. As such, Ki-67 has been suggested to have value as an important biomarker in the evaluation of cancer prognosis, with potential predictive significance in the treatment of cancer as well [11,12]. Nevertheless, the variety of Ki-67 assays and lack of standardization has resulted in inconsistent inter-observer and inter-laboratory reproducibility, limiting the universal application of Ki-67 for prognostic and predictive evaluation in breast cancer in routine clinical practice [2,13].

Although there *is* a collective recognition among the international guideline recommendations that Ki-67 is a prognostic biomarker in breast cancer, the international guideline recommendations for the use of Ki-67 in the prognostic and predictive evaluation of breast cancer remains mixed [14]. Although several international organizations, including the Italian Association of Medical Oncology [15], European Group on Tumor Markers [16], European Society for Medical Oncology [17], and National Institute for Health and Care Excellence [18] either recommend or consider Ki-67 for use in the prognostic evaluation of breast cancer patients [14], neither the American Society of Clinical Oncology [19,20,21,22] or the National Comprehensive Cancer Network [23] support the use of Ki-67 in the prognostic or predictive evaluation of breast carcinoma. Only the European Society for Medical Oncology supports the use of Ki-67 expression as predictive of response to systemic chemotherapy, and only in the neoadjuvant setting [24]. The American Joint Committee on Cancer does not give any specific recommendations on the use of Ki-67 [14]; however, they recommend that “providers and registries should continue to collect and record Ki-67 results” [25].

The future of Ki-67 in this era of precision cancer therapy will depend on accurate and reproducible reporting of Ki-67 immunohistochemistry, only achievable through standardization of pre-analytic, analytic, and post-analytic variables. The recent FDA approval of abemaciclib (based on the seminal monarchE trial [26]), a cyclin-dependent kinase (CDK) 4/6 inhibitor, as an adjuvant treatment option for high-risk ER-positive/human epidermal growth factor receptor-2 (HER2)-negative early stage breast cancer, relies on a quantitative assessment of Ki-67 expression in the treatment decision algorithm. This further reinforces the urgent need for standardization of Ki-67 antibody selection and Ki-67 staining interpretation, with validated cutoffs for low and high Ki-67 expression, ultimately validated clinically against patient outcomes. Such standardization would likely lead to international guideline consensus recommendations on the use of Ki-67 as a prognostic and predictive marker in breast cancer. 

The goals of this review will be to highlight the historical evolution of Ki-67 in breast cancer, summarize the present literature, including the International Ki-67 in Breast Cancer Working Group (IKWG) efforts to standardize Ki-67 assessment [2,13], with an emphasis on the clinically-relevant literature, and to consider the future implications of the monarchE trial (which led to the U.S. approval of Ki-67 as a companion diagnostic biomarker in breast cancer) [26], and other recent developments in breast cancer Ki-67 research. Furthermore, we will discuss the necessary changes required across pathology practices to help increase the reliability and widespread adoption of Ki-67 as a prognostic and predictive marker for breast cancer in clinical practice.

## 2. Ki-67—Past Observations

### 2.1. The Discovery of Ki-67 

In 1983, Dr. Johannes Gerdes published what should be considered a groundbreaking article, introducing a mouse monoclonal antibody that recognized a nuclear antigen that was present only in proliferating cells [27]. This antibody, produced in Kiel, Germany, and cloned from the 67th well of a multititer plate [28], was given the name “Ki-67” (pronounced as “kē-67). After establishing that the expression of the Ki-67 antibody correlated with tumor growth fraction, Dr. Gerdes reported that the Ki-67 antigen was expressed in all cell cycle phases, with the exception of quiescence (G_0_) and the early G_1_ phase, when the cells enter the cell cycle for the first time after a resting state [9]. Initially, Dr. Gerdes suggested that Ki-67 could be of prognostic value in non-Hodgkin‘s lymphomas [9]. However, the observation that Ki-67 expression was only present in proliferating cells suggested to Dr. Gerdes that Ki-67 might be a useful tool for prognostic evaluation in carcinoma as well. Several years later, it was shown that there are subnuclear levels of the Ki-67 antigen with very low levels of expression in the quiescent (G_0_) and the early G_1_ phase; however, this expression is relatively insignificant and is not detectable by standard immunohistochemistry methodologies [10].

There are nearly 30,000 base pairs within the human genome that encode for the Ki-67 protein [29,30]. Although the function of Ki-67 is still a matter of discussion, the evolving literature suggests the Ki-67 protein is associated with the maintenance of mitotic chromosomes, via chromosome dispersion and chromosome clustering, potentially through a surfactant-like mechanism at the boundary between mitotic chromatin and the cytoplasm [31,32]. Ki-67 appears to be a peri-chromosomal nuclear protein [33] with a complex primary structure characterized by 10 putative nuclear targeting sequences, 143 potential phosphorylation sites for protein kinase C, 89 sites for casein kinase II, 2 tyrosine kinase sites, and 8 consensus sites for cdc2 kinase [34]. The central part of the Ki-67 antigen contains 16 tandemly repeated 366-bp elements, the “Ki-67 repeats”, each including a highly conserved new motif of 66 bp, the “Ki-67 motif”, which encodes for the epitope detected by the Ki-67 antibody [34]. A common feature for a number of proteins involved in cell cycle regulation is the so called forkhead-associated (FHA) domain, which is also found to be present in the sequence motifs of the Ki-67 protein [35], and there is evidence supporting that the Ki-67 protein is a requirement for cell proliferation [31,32,34,36].

### 2.2. The Development of Ki-67 Immunohistochemistry for Clinical Practice

Initially, the use of Ki-67 clinically was hampered by the fact the evaluation of Ki-67 could only be performed on frozen tissue. The vast majority of mouse monoclonal antibodies generated prior to 1990 did not appreciably stain in formalin-fixed paraffin-embedded (FFPE) tissue sections. As such, prior to 1990, mouse monoclonal antibodies were limited to use on frozen tissue sections or fresh tissue smears. In 1990, the first mouse monoclonal antibody to be detected in FFPE tissue was immunolocalized and reported, recognizing the proliferating cell nuclear antigen (PCNA) [37]. The immunolocalization of Ki-67 was not possible at that time, due to the complex structure of *MKI67* (the Ki-67 gene); however, attempts to clone *MKI67* were finally successful [34], and novel anti-Ki-67 monoclonal antibodies raised against recombinant protein expressed in bacteria were subsequently produced. The most clinically important of these novel anti Ki-67 monoclonal antibodies were named MIB1 and MIB3 (Molecular Immunology Borstel 1 and 3, named after the laboratory in Borstel Germany, where it was first produced) [38]. Both MIB1 and MIB3 recognized the Ki-67 antigen epitope encoded by the Ki-67 motif, and most importantly, through the use of a new protocol for immunohistochemical staining using antigen retrieval [39], both MIB1 and MIB3 allowed for the staining of FFPE tissue [38]. Since those landmark 1992 [38] and 1993 [34] studies, the MIB1 monoclonal antibody against the Ki-67 antigen has been universally used for almost 30 years to determine the proliferation characteristics of hematopoietic malignancies and solid tumors, including breast cancer.

## 3. Ki-67—Present Directions

Most of the clinically-oriented research on Ki-67 in breast cancer to date can be divided into five main categories: (1) the use of Ki-67 as a prognostic marker in breast cancer overall; (2) differences in Ki-67 expression among breast cancer molecular subtypes; (3) the utility of Ki-67 as a prognostic and predictive marker in the adjuvant setting; (4) the utility of Ki-67 as a prognostic and predictive marker in the neoadjuvant setting; and (5) efforts by the International Ki-67 in Breast Cancer Working Group (IKWG) to standardize and improve the accuracy, reproducibility, and validity of Ki-67 assessment in breast cancer.

### 3.1. The Prognostic Value of Ki-67 in Breast Cancer

In 1986, Dr. Gerdes published the first paper on the immunohistochemical determination of the growth fraction with Ki-67 in breast tissue, including breast neoplasms. The paper clearly showed that the expression of Ki-67 was significantly associated with increasing grade in breast carcinoma [5]. Since that seminal study, over 6000 articles addressing the use of Ki-67 in breast carcinoma have been published (PubMed database November 2022). 

Proliferative activity, in general, has been shown to be a significant factor in determining the prognosis of breast cancer [1,40,41,42], and the monoclonal Ki-67 antibody has been consistently shown to be an independent prognostic biomarker in breast cancer patients [43], with multiple studies showing a correlation between the Ki-67 proliferation index and breast cancer survival [2,43,44,45,46,47,48,49]. However, since the establishment of Ki-67 as a prognostic variable in breast cancer, the more pertinent question from a clinical perspective has become the need to better define cutoff points along the Ki-67 expression continuum, such that breast cancer patients can be more consistently and accurately stratified into groups with higher and lower risk of recurrence as well as those more and less likely to benefit from systemic treatments. Several studies have attempted to address this issue, with various methodologies used to determine a “specific” Ki-67 cutoff point, and different cutoff points selected depending on the clinical context.

In a 2002 meta-analysis by Mirza et al. [44], five studies including 1959 patients with breast cancer were evaluated based on Ki-67 cutoff points ranging from 1% to 25%. By univariate analysis, Ki-67 was a significant prognostic factor for overall survival (OS) and/or disease-free survival (DFS) in three of five studies and by multivariate analysis, Ki-67 was a significant prognostic factor in four of five studies for OS and/or DFS. In a 2007 meta-analysis by de Azambuja et al. [45], 46 studies including 12,155 patients with breast cancer and Ki-67 data were evaluated based on Ki-67 cutoff points ranging from 3.5% to 34%. Ki-67 positivity was associated with higher probability of relapse and worse survival in both node-negative patients (*p* < 0.001) and node-positive patients (*p* < 0.001). Similarly, in a 2008 meta-analysis by Stuart-Harris et al. [50], forty-three studies including 15,790 patients with breast cancer and Ki-67 data were evaluated based on Ki-67 cutoff points ranging from 0% to 28.6%. In this study, tumors overexpressing Ki-67 were associated with significantly shorter OS and DFS, using results from univariate and multivariate analyses from the individual studies, although there was “evidence of significant study heterogeneity” [50]. However, these and other earlier studies [46,47,48,49] were limited by variability in Ki-67 antibody selection, assay interpretation, and retrospective study design. Although these early meta-analyses reinforced the prognostic potential of Ki-67, they also reinforced the obstacles preventing widespread adoption of Ki-67 as a prognostic marker for breast cancer in clinical practice. The results of these meta-analyses are summarized in Table 1.

A 2013 study by Inwald et al. [51] evaluated 3658 patients diagnosed with breast cancer between 2005 and 2011 with available Ki-67 data, identified from a clinical cancer registry in Bavaria, Germany. In this study, clear criteria were outlined for the evaluation of Ki-67, following the published guidelines by the then new IKWG [13], using the accepted “gold standard” Ki-67 monoclonal antibody MIB1, with evaluation of whole tumor sections, not limited to hot spots, and using a Ki-67 cutoff of 15% as defined by national and international recommendations at the time. Ki-67 values were categorized into five distinct categories (≤15%, 16–25%, 26–35%, 36–45%, and >45%) for survival analyses. A strong correlation was found between the Nottingham grade and Ki-67 (*p* < 0.001), and in multivariable analysis, Ki-67 was an independent prognostic parameter both for DFS in patients with a Ki-67 > 45% (*p* = 0.001), as well as for OS in patients with a Ki-67 > 25% (*p* < 0.05), independent of common clinical and histopathological factors. In this study, the 5-year DFS rate was 86.7% in patients with Ki-67 expression ≤15% compared to 75.8% in patients with Ki-67 expression >45%. The 5-year OS rate was 89.3% in patients with Ki-67 expression ≤15% compared to 82.8% in patients with Ki-67 expression >45%.

### 3.2. Ki-67 in Breast Cancer Molecular Subtypes

It is well established that breast carcinomas can be subclassified into four major molecular subtypes: luminal A, luminal B, HER2-enriched, and basal-like (triple-negative) carcinomas [52,53]. As a result, substantial research has been devoted to determining the differences in Ki-67 expression across molecular subtypes. Cheang et al. [54] reported that luminal A and luminal B subtype cancers can be separated with a Ki-67 cutoff of 14%, with luminal A subtypes having a Ki-67 <14%, and luminal B subtypes having a Ki-67 of ≥14%. In this study, the cutoff of 14% was determined using receiver-operating characteristic analyses with a gene expression profile-defined gold standard. The Cheang et al. [54] defined cutoff point of 14% to define luminal A and luminal B subtypes was initially supported by the St. Gallen 2011 consensus [55], which subsequently in 2013 added a PR status of <20% or ≥20% as an alternative variable to distinguish the luminal B and luminal A subtypes [56], respectively. However, the St. Gallen 2015 consensus subsequently abandoned a specific cutoff point for Ki-67 due to analytic and pre-analytic barriers to standardized assessment, as it was felt that “while high and low values are reproducible and clinically useful, there appears to be no optimal [Ki-67] cut point…”, and that “Ki-67 scores should be interpreted in the light of local laboratory [median] values…” [57]. Although the identification of a single useful cutoff point to define a low and high Ki-67 in all breast cancer patients may not be achievable, the identification of reproducible cutoffs with prognostic and predictive value in specific clinical contexts remains a worthy and active area of investigation.

Outcomes-based studies have confirmed the importance of Ki-67 as a prognostic marker in luminal breast cancers, similar to the prognostic differences seen in luminal A and luminal B disease as identified by molecular gene-expression panels [58,59,60,61]. In addition, Ki-67 appears to have prognostic significance within luminal B disease. For instance, Péréz-López et al. [62] showed that Ki-67 was associated with poor survival in patients with node-negative luminal B breast cancer. Similarly, Criscitiello et al. [63] reported that a cutoff point of 32% for Ki-67 expression could identify a subset of patients with luminal B and node-positive breast cancer who could benefit from the addition of adjuvant chemotherapy to endocrine therapy. 

Higher levels of Ki-67 have been reported in HER2-positive and triple-negative breast cancers compared with HER2-negative luminal tumors, particularly compared to luminal A disease [64,65,66]. BRCA1-mutated triple-negative tumors appear to have a particularly high Ki-67 proliferative index, often exceeding 60% [40,67]. In one study by Vaile et al. [46], HER2 overexpression was associated with high Ki-67 expression in postmenopausal patients, but not in premenopausal patients, highlighting, as previously mentioned [68], the need to take into account the clinical context, including hormone expression, menopausal status, and molecular subtypes when considering the expression of Ki-67. More recent data suggest that HER2-low breast cancers may have a significantly lower Ki-67 than both HER2-positive and HER2 negative (score 0) tumors [69]. Unlike in luminal breast cancer, Ki-67 has been reported by some to have no significant association with survival in HER2-positive and triple-negative tumors [68,70,71], although some authors have suggested that the lack of an observed association might be more due to use of lower cutoffs (such as 15%) that are more appropriate for luminal tumors [70]. Indeed, some studies with higher cutoffs (such as 30% or 40%) have shown a significant association with prognosis in triple-negative breast cancer [72,73,74], although this finding has not been replicated in all studies [75]. Ki-67 has also been associated with higher expression of PD-L1 in several studies [76,77,78,79], suggesting the possibility of a predictive role for Ki-67 in selecting patients for immune checkpoint inhibitor therapy.

### 3.3. Ki-67 as a Prognostic and Predictive Marker in the Adjuvant Setting

There has been extensive research to date studying the potential for Ki-67 to serve as both a prognostic and predictive biomarker in the setting of adjuvant endocrine therapy and chemotherapy for breast cancer patients. For instance, in a prospective randomized trial evaluating the prognostic and predictive value of Ki-67 comparing letrozole with tamoxifen as adjuvant endocrine therapy in postmenopausal women with early breast cancer, Viale et al. [80] examined a Ki-67 cutoff point of ≤11 and >11 in 4399 patients. Specific criteria were established for number of cells counted (definite nuclear immunoreactivity in 2000 invasive neoplastic cells in randomly selected high-power (×400) fields at the periphery of the tumor), and the Ki-67 cutoff point of 11% was established based on the median value of Ki-67 in the frequency distributions of the Ki-67 labeling index in the trial cohort. In this study, higher values of Ki-67 expression were associated with worse DFS, and a treatment benefit was associated with letrozole compared to tamoxifen in patients with a high Ki-67. 

Most studies have found that high Ki-67 values have been associated with worse prognosis, but the data regarding the ability of high Ki-67 to predict sensitivity to adjuvant chemotherapy have been more mixed [46,49,58,63,80,81,82,83]. One study by Viale et al. [46] examined Ki-67 in 1924 patients enrolled in two randomized International Breast Cancer Study Group trials of adjuvant chemoendocrine therapy vs. endocrine therapy alone for node-negative breast cancer. In this study, a Ki-67 cutoff point of 19% was established based on the median value of Ki-67 in the frequency distributions of the Ki-67 labeling index in these two trial cohorts. This study also reported that a high Ki-67 expression was associated with worse DFS; however, high Ki-67 expression was associated with worse DFS in patients in both cohorts (patients who received chemoendocrine therapy and patients who only received endocrine therapy), suggesting that Ki-67 expression was *prognostic*, but not *predictive* of any benefit with chemoendocrine therapy compared with endocrine therapy alone. By contrast, as previously mentioned, in a propensity score-matched retrospective cohort study of 1241 patients with luminal B node-positive breast cancer, Criscitiello et al. demonstrated a significant interaction effect for benefit from addition of chemotherapy in patients with a high Ki-67, defined as ≥32% [63]. The potential for Ki-67 to serve as a predictive marker in this clinical context is also indirectly supported by research showing that Ki-67 expression has moderate to good concordance with multigene expression panels (such as Oncotype DX and MammaPrint) designed to aid clinicians in deciding whether to use adjuvant chemotherapy in hormone-receptor breast cancer patients (see Section 4.1 below) [84,85,86].

The potential clinical utility of Ki-67 in the adjuvant setting is greater than ever with the recent publication of the results of the monarchE randomized clinical trial [26,87,88]. The monarchE trial [26,87] randomly assigned 5637 high risk patients—defined as patients with four or more positive nodes, or one to three nodes and either tumor size ≥5 cm, histologic grade 3, or central Ki-67 ≥20%—to either standard-of-care adjuvant endocrine therapy alone (*n* = 2829), or standard-of-care adjuvant endocrine therapy plus the CDK 4/6 inhibitor abemaciclib (*n* = 2808). In this study, Ki-67 was centrally assessed and confirmed, using a standardized assay and methodology, developed and validated for sensitivity, specificity, repeatability, precision, and robustness, using detailed scoring instructions and a predefined cutoff of 20% (<20%, ≥20%) [26,87,88]. The Ki-67 threshold of ≥20% was chosen based on recommendations from the St. Gallen International Breast Cancer 2015 conference [57,87]. Pathologists examined the whole slide and chose representative areas to count, including hotspots and non-hotspots, and then calculated the number of Ki-67 positive viable tumor cells divided by the total number of viable tumor cells over the whole tumor area. A Ki-67 level of ≥20% was prognostic for an inferior outcome; however, an adjuvant abemaciclib benefit was seen regardless of a high (≥20%) or low Ki-67 (<20%) expression, although the magnitude of the benefit was higher in the Ki-67 high group of patients [26,87]. It remains to be determined whether Ki-67 values may be more predictive of benefit in patients with limited nodal disease but otherwise lower-risk pathologic features.

### 3.4. Ki-67 as a Prognostic and Predictive Marker in the Neoadjuvant Setting

Ki-67 has also shown utility as a prognostic and predictive biomarker in the neoadjuvant setting. Similar to hormone receptors and HER2, changes in Ki-67 have been reported following neoadjuvant chemotherapy, with reports suggesting that about two thirds of cases showing a decrease in Ki-67, one fourth of cases showing an increase, and the rest showing no change, with exact percentages varying depending on the distribution of breast cancer subtypes and type of neoadjuvant chemotherapy being used [89,90,91,92]. This change makes intuitive sense, as one would expect the most proliferative tumor cells to generally be most vulnerable to cytotoxic chemotherapy, which generally targets proliferating cells. High values of Ki-67 following chemotherapy, as well as a lack of reduction in the Ki-67 value, have been found by many authors to be associated with worse prognosis [93,94,95,96,97,98,99,100,101,102], likely reflecting tumors that are aggressive and relatively insensitive to systemic chemotherapy. Some authors have suggested a more nuanced interpretation, with reductions in Ki-67 associated with better prognosis for luminal B, HER2-positive, and triple-negative, but not luminal A tumors [103]. 

Researchers have demonstrated the predictive and prognostic potential of Ki-67 prior to chemotherapy administration, with most studies suggesting that higher pre-treatment Ki-67 values are associated with a greater probability of pathologic complete response and worse survival [93,99,101,104,105]. In a 2013 study involving 1166 patients in the neoadjuvant GeparTrio trial, Denkert et al. [68] reported significant pathologic complete response (pCR), DFS, and OS over a wide range of Ki-67 cut points (3–94% for pCR, 6–46% for DFS, and 4–58% for OS); however, Ki-67 performed differently across molecular subtypes if analyzed as a predictive marker compared with analysis as a prognostic marker [68]. In this study, the predictive power of Ki-67 was lower in the HER2-positive subtype and only added prognostic information in hormone receptor-positive tumors, in which a lower Ki-67 was associated with better DFS and OS. Ki-67 was not significantly associated with DFS and OS in triple-negative tumors. Furthermore, the study results suggested that increased expression of Ki-67 was an indicator of poor prognosis in patients who are not responding to chemotherapy over a wide range of cut points; however, it was potentially an indicator of *good* prognosis in subsets of patients with a Ki-67 between 20% and 30% who responded to chemotherapy. These results clearly highlighted the potential limitations in using random cutoff points when using Ki-67 as a prognostic marker in the evaluation of breast cancer; however, given the clear relationship of Ki-67 with DFS and OS in hormone receptor-positive patients, the authors suggest that there is clear value in using Ki-67 expression as a continuous variable reflecting the percentage of proliferating cells in the tumor. The authors recommended caution in making clinical decisions for values that are very close to any given selected cutoff point and reinforced that the performance of Ki-67 when using cutoff points is likely dependent on the clinical end points, the response to therapy, and the molecular subtypes. Similarly, the more contemporary literature continues to reinforce the prognostic value of Ki-67 expressed as a continuous variable [2,40,84,106,107,108,109,110].

In addition to its prognostic role for neoadjuvant chemotherapy, reductions in Ki-67 in serial biopsies during neoadjuvant endocrine therapy treatment have been associated with improved outcomes and can be used as a surrogate for endocrine response [2,111]. A low Ki-67 has also been suggested to be an indicator of whether or not additional treatments after surgery are justified in patients who received neoadjuvant endocrine therapy treatment [2,112,113]. Several other studies have evaluated Ki-67 expression in an effort to assess patient response during or after neoadjuvant systemic chemotherapy [2,114,115,116,117,118,119,120,121,122,123,124,125], generally with a strong correlation between Ki-67 expression and long term outcomes after neoadjuvant chemotherapy is completed. The IKWG acknowledges that Ki-67 may be of value in decisions regarding use of neoadjuvant endocrine or systemic chemotherapy, based on the expression of Ki-67 and the presence or absence of residual disease [2]. However, early reductions of Ki-67 in the early phases of neoadjuvant systemic chemotherapy [2,126], and the lack of reduction in Ki-67 in serial biopsies during neoadjuvant endocrine therapy treatment in which patients were subsequently escalated to neoadjuvant systemic chemotherapy [2,112], have not been sufficiently shown to be associated with a significantly high enough rate of pathological complete response after chemotherapy to support a convincing recommendation by the IKWG for a predictive effect of Ki-67 with any type of neoadjuvant therapy. Additional studies addressing this issue are in progress [2,127], and at the current time, the IKWG considers the use of Ki-67 in the neoadjuvant setting to be only investigational and does not recommend using Ki-67 to optimize treatment for individuals receiving either endocrine or systemic neoadjuvant therapy [2].

### 3.5. Recommendations of the International Ki-67 in Breast Cancer Working Group (IKWG)

The current international recommendations on the clinical utility of Ki-67 in breast cancer are mixed [14]. Over the last 10 years, there has been a significant focus on the work done by the International Ki-67 in Breast Cancer Working Group (IKWG), in an effort to establish accepted universal guidelines for Ki-67 antibody selection, staining protocol, and staining interpretation [2]. In 2011, the IKWG published original guidelines in breast cancer on the pre-analytical and analytical evaluation, as well as Ki-67 interpretation, scoring and data handling. The consensus group recognized, however, that scoring procedures varied, and the lack of standardization for scoring Ki-67 expression in different types of specimens (i.e., biopsy vs. whole-tumor sections vs. TMAs) was problematic. Moreover, the consensus group recognized the importance of establishing quality assurance protocols to ensure standardization of Ki-67 analysis across different laboratories. Without these standards in place, the 2011 IKWG felt that application of specific cutoffs for decision making “must be considered unreliable unless analyses are conducted in a highly experienced laboratory with its own reference data.” 

In 2019, the IKWG published updated guidelines regarding the analytical validity and clinical utility of Ki-67 in the clinical care of breast cancer patients, recommending that Ki-67 only be used in a single clinical setting if intra-laboratory analytic validity can be achieved [2]. In addition, when comparing different laboratories, Ki-67 staining protocols should be evaluated for consistent, inter-laboratory reproducibility, coupled with existing external quality assurance protocols. The 2019 IKWG undertook a series of multi-institutional studies designed to further assess the reproducibility of Ki-67 [2,128,129,130,131,132]. An important finding was that although there were pre-analytical and staining issues that could be improved upon, the most significant issue affecting reproducibility was the variability in Ki-67 scoring between different laboratories [128]. When a standardized counting methodology was introduced, inter-observer variability was reduced [129]. Based on these observations, the IKWG published the recommended criteria outlined in Table 2 for analytical validity when developing a methodology for evaluating Ki-67 expression [2].

Although these recommendations have value for achieving acceptable intra-laboratory analytic validity, the 2019 IKWG still recognizes that there are multiple sources of inter-laboratory variation due to various pre-analytic and analytic differences, particularly the interpretation of Ki-67 scoring between laboratories (Table 3) [2]. A recent study by the IKWG found that Ki-67 expression scores were higher for core biopsy slides compared to paired whole sections from resections (*p* ≤ 0.001) [133], likely due to pre-analytical factors such as tissue handling and fixation. Although the IKWG considers testing on core biopsy and excision specimens to both be suitable, the IKWG considers the core biopsy to best reflect the biological status of the tumor and prefers testing Ki-67 on the core biopsy specimen. IKWG also recommends following American Society of Clinical Oncology/College of American Pathologists (ASCO/CAP) guidelines for breast tissue handling, such as the use of internal on-slide and batch-to-batch controls (positive and negative) [2,20,134,135], avoidance of prolonged storage of cut section at room temperature, as well as high-temperature antigen retrieval. Quality assurance and control should be established and maintained in each laboratory with participation in mandatory quantitative external quality assessment. The IKWG recommends counting all positive staining cells, regardless of intensity (Figure 1A–H), and requires counting up to 400 invasive carcinoma cells in up to four separate fields using a weighted global scoring methodology (Figure 2A,B), which attempts to account for heterogeneity of Ki-67 expression (Figure 3A,B). The IKWG also studied a “global” vs. “hot spot” counting approach when evaluating cells with positive Ki-67 expression. The hot spot approach was associated with higher inter-observer variability and with more reproducibility using the global counting approach; however, differences were not statistically significant. The IKWG supports considerations for digital imaging, although additional studies are needed to recommend this approach. Finally, the IKWG recommends capturing Ki-67 data as a continuous percentage variable rather than using different groups with specific cutoff points.

In addition to assuring both intra-laboratory and inter-laboratory reproducibility, laboratories must agree to participate in the central assessment of Ki-67 results from multiple laboratories, the process of “external quality assessment” or EQA, so that each laboratory can benchmark their results against those of their peers. Currently, the CAP and United Kingdom are in the planning and implementation stages for development of such a comprehensive EQA protocol for Ki-67 [2], which will potentially have a major impact on achieving more widespread use of Ki-67 in the clinical management of breast cancer patients. 

In summary, the IKWG does acknowledge that Ki-67 has clinical validity for the determination of prognosis in patients with ER-positive early stage breast cancer [2]; however, the IKWG only supports a limited clinical utility for Ki-67 in the clinical management of ER-positive breast cancer patients, and only if an analytically validated assay and scoring system is used. There is insufficient evidence to support an IKWG recommendation for the use of Ki-67 in ER-negative breast cancer patients. The IKWG support for Ki-67 clinical utility is limited to eliminating the use of gene expression assays in women with ER-positive breast cancers that have a favorable anatomic prognosis, if the Ki-67 levels are ≤5% or ≥30%. The 5% and 30% cutoffs were chosen because they are sufficiently different from clinically-relevant prognostic decision cutoffs to not be impeded by residual analytical variability after assay standardization [136]. Thus, the IKWG only supports the use of a Ki-67 of 5% or less being considered “low”, and a Ki-67 expression of 30% or higher being considered “high,” and supports withholding chemotherapy for a low Ki-67 or proceeding with chemotherapy for a high Ki-67, without the need for more expensive commercial gene expression assays [2]. This should of course be considered in coordination with other histopathological variables and the patient’s clinical history, and the IKWG does not recommend the use of Ki-67 to withhold chemotherapy based on a low Ki-67 in patients with a poor anatomic prognosis (i.e., positive nodes, large tumor size, etc.), even if they have a favorable biology (i.e., ER rich, HER2 negative) [2].

## 4. Ki-67—Future Considerations

The future application of a Ki-67 threshold for clinical decision making in the management of high risk ER-positive breast cancer in current pathology laboratories will depend on the standardization of pre-analytic, analytic, and post-analytic factors involved with Ki-67 testing. This is not an unreachable goal, as improvements in standardization of pre-analytical factors, analytical performance, and participation in proficiency testing programs led to improvements in the analytic performance and diagnostic accuracy of ER, PR, and HER2 as biomarkers in the management of women with breast cancer, with international consensus as to their clinical utility [137,138,139,140]. There are several important potential future research applications to help improve the standardization and clinical utility of the Ki-67 assay.

### 4.1. Potentially Cost-Effective Alternative to Genomic Profiling Assays

Oncotype DX^®^ (ODX) (Genomic Health, Redwood City, CA, USA) is a 21 gene commercial reverse transcription quantitative real-time PCR (RT-qPCR) mRNA-based multigene assay developed for use with FFPE breast cancer patient tissue that is ER-positive and lymph node-negative [6,14,141]. The ODX test uses an algorithm to calculate a recurrence score (RS), which is reported as either low risk, intermediate risk, or high risk, and is heavily influenced by five proliferation-related genes, one of which is *MKI67* [6,14]. Ki-67 as assessed by immunohistochemistry has been shown to have a strong correlation between the Ki-67 expression score and the ODX RS, with the strongest correlations apparently being found in the higher-Ki 67/higher RS groups [142,143,144,145]. Standardization of Ki-67 assays will increase the clinical utility of Ki-67 as an adjunct to ODX in decisions on adjuvant chemotherapy in hormone receptor-positive, HER2 negative, lymph node-negative breast cancer patients [20].

Ki-67 has been used as part of an immunohistochemical approach for assessing clinical risk in breast cancer [86,146,147,148,149,150,151,152,153]. Standardization of the Ki-67 antibody as well as standardization of staining methodologies will further increase the value of Ki-67 using these immunohistochemical methodologies. Risk-stratification models such as the IHC4 score [86,146,147,148,149] and the Magee equations ^TM^ [150,151,152,153], both use semi-quantitative information from the immunohistochemical assessment of Ki-67 and have been validated in retrospective studies as being able to identify patients at lower or higher risk of breast cancer relapse following endocrine therapy [148,149,150,151,152,153]. Algorithmic approaches using either the new Magee score ^TM^ [152,153] or the average Modified Magee score [150,151] have identified specific cutoff points which identify patients at lower or higher risk of breast cancer relapse following endocrine therapy. Turner et al. [151] published the first study showing an association between the Magee score ^TM^ (average modified Magee score) and clinical outcomes, suggesting that patients with an average modified Magee score of ≤12 had a comparably low risk of breast cancer recurrence when compared to the low-risk categories defined by ODX, in an average follow-up of 79 months (6.6 years). However, all of these immunohistochemical approaches using Ki-67 suffer from the previously discussed lack of standardization of pre-analytic, analytic, and post-analytic factors involved with Ki-67 testing. The encouraging results that have been seen with immunohistochemical risk-stratification models to date suggests that further research to improve the standardization of Ki-67 would likely lead to improvements in the performance of these models, as well.

### 4.2. Potentially Routine Companion Diagnostic Test in Breast Cancer

Based on the data from the monarchE clinical trial, the FDA has approved abemaciclib in combination with endocrine therapy for the adjuvant treatment of hormone receptor-positive/HER2-negative, node positive, early breast cancer with a Ki-67 score ≥20%. Of note, the FDA-approval specifically requires that the Ki-67 score be determined by the U.S. FDA-approved Ki-67 IHC MIB-1 pharmDx (Dako Omnis, Santa Clara, CA, USA) assay as a companion diagnostic test. This assay is now the first companion IHC assay to identify patients with early breast cancer at high risk of disease recurrence, for whom adjuvant treatment with abemaciclib in combination with endocrine therapy is being considered. This approval would have a significant impact on the pathology practice, especially in those pathology laboratories which do not have the DAKO platform, as they would need to expend significant resources in order to acquire the platform and validate the test, even if they have already validated another Ki-67 assay using a different platform.

There has been some controversy regarding the requirement of the companion Ki-67 assay as part of the FDA approval of abemaciclib. The analytic and clinical validity of the Ki-67 results from the monarchE clinical trial was strengthened by the use of a centrally confirmed standardized Ki-67 assay. According to the FDA Summary of Safety and Effectiveness Data (SSED), the reproducibility of the Ki-67 assay was determined by having 60 pre-stained sections interpreted by three different trained pathologists, each scoring three separate times, with between-observer overall agreement reported at 97.2% and within-observer overall agreement reported at 98.1% for the ≥20% criteria [154]. Although this result met the pre-specified study acceptance criteria, some authors have suggested that this type of study is insufficient to determine the true level of between- and within-observer variability likely to occur in real practice, with some studies on other biomarkers suggesting that reproducibility is substantially worse when tested on more than eight pathologists compared to only two or three pathologists [155,156,157]. The standardized Ki-67 methodology was further validated by Polewski et al. [88] in 1954 patients from the monarchE clinical trial who received endocrine therapy alone, with Ki-67 reproducibility studies (inter-laboratory, intra-laboratory, inter-observer, and intra-observer) conducted at three external laboratories, again using detailed scoring instructions. This study also showed good within- and between-pathologist concordance, with >90% negative, positive, and overall agreement observed for all tested comparisons [88]. However, they also reported that one pathologist scored about 7–8% lower in one run compared to the other two runs of the study (Supplemental Figure 6 in Polewski et al. [88]), suggesting at least the possibility of individual scoring errors that could lead to clinically meaningful differences in interpretation, even if overall agreement is good. In this study, however, patients with a Ki-67 ≥20% had a clinically meaningful increased risk of developing invasive disease within 2 years compared with those with Ki-67 <20%, further validating the prognostic value of Ki-67 at this specific cutoff point.

Although invasive disease free survival (IDFS) was significantly better in both the intention-to-treat (ITT) cohort and the pre-specified subgroups with Ki-67 ≥20%, the FDA required the use of the Ki-67 companion diagnostic assay as part of the approval because of incomplete overall survival data, suggesting that there was a possibility of harm (i.e., potential detriment) in the ITT cohort (HR = 1.091, 95% CI 0.818–1.455) but not in the cohort with node positivity and high-risk pathologic features, including Ki-67 ≥20% (HR = 0.767, 95% CI 0.511–1.152) [158]. Some have questioned this decision [157,159], arguing that Ki-67 was merely acting as a prognostic marker, not a predictive marker, as the hazard ratios for IDFS were not significantly different in the Ki-67 ≥20% (HR = 0.63, 95% CI 0.49–0.80) and Ki-67 <20% (HR = 0.70, 95% CI 0.51–0.98) subgroups, among the subset of patients who were included in the study regardless of Ki-67 status [87]. In fact, the European Medicines Agency (EMA) has not required the use of Ki-67 as a companion diagnostic test with abemaciclib. However, it is unclear whether the study was powered to detect a significant interaction effect in IDFS and the question of what predictive role a high Ki-67 might have in patients without other high-risk pathologic features (i.e., grade 3 disease or size ≥ 5 cm) but were otherwise eligible for trial inclusion remains unanswered. In any case, it seems reasonable to expect that the FDA-approved indication for abemaciclib will continue to evolve as the overall survival data from the monarchE trial continue to mature.

Despite these controversies, for the time being, the results of the monarchE clinical trial potentially opens the door for the consideration of Ki-67 as a routine companion diagnostic test—similar to ER, PR and HER2—for all breast cancers. This trial also importantly demonstrated that the Ki-67 immunohistochemical assay, when scored in a rigorous fashion, can be reliable at cutoffs below the ≥30% cutoff recommended by the IKWG [2]. Further research regarding the reliability of different scoring methods at the ≥20% cutoff, as well as additional research to further validate the clinical utility of this cutoff, is therefore warranted. 

### 4.3. Simplified Scoring Algorithms

Although the IKWG scoring method for Ki-67 fulfils their published criteria for analytic validity, the IKWG admits that this scoring method is tedious. The IKWG scoring method requires both installation of software and pathologist training. The median scoring time per case is 9 min [2]. The IKWG admits that the “level of attention to training, and time required to perform each assay, may be challenging to achieve in the routine pathology laboratory setting,” and that, “the clinically relevant Ki-67 index range of 10–20% could still cause misassignment of some cases” [2]. Simpler scoring methods have been proposed based on quick visual estimates of key cutoffs defined by easy-to-recognize ratios of positive–negative cells, such as the “Eye-5” and “Eye-10” methods or the use of standard reference cards, and such methods may allow for reproducible grouping of cases into clinically relevant categories [160,161,162,163]. However, these simpler methods for assessing Ki-67 have so far received less attention from the IKWG compared to their previously described manual counting and weighted average method [2], which undoubtedly better reflects the biological variability of breast cancer but may not be strictly necessary to divide patients into clinically useful categories. Automated approaches to Ki-67 scoring are less tedious but present their own challenges (see Section 4.4 below), limiting their potential usefulness and widespread adoption in routine pathology practice in the short term. Thus, there remains a need for additional research evaluating standardized scoring methods that can be analytically validated but are also more practical for use in routine pathology practice.

### 4.4. Automated Digital Image Analysis

Numerous studies have demonstrated the potential for digital image analysis to provide accurate automated scoring of Ki-67 [64,107,144,164,165,166,167,168,169,170,171,172,173,174,175,176,177]. Although a full discussion of the research on automated digital image analysis of Ki-67 in breast cancer is beyond the scope of this review, the studies to date have generally shown that automated methods typically match the accuracy of manual estimates using rigorous standardized scoring methods (such as that proposed by the IKWG) and exceed the accuracy of visual estimates using less standardized methods [2,174,176,178]. Automated digital image analysis therefore offers the potential benefit of assay standardization and a high degree of accuracy and reproducibility, without the downside of standardized manual scoring methods involving counting large numbers of tumor cells, which may be difficult to implement in routine clinical practice [157,176]. 

Although much research has been done on the potential benefits of automated digital image analysis for improving the reliability of Ki-67 analysis assay interpretation, less research has been done on the processes for actually validating and implementing these algorithms into a routine clinical workflow. Whole slide scanning, typically a pre-requisite for automated digital image analysis, requires significant upfront investments and can present substantial logistical challenges [179]. Whole slide scanning has not yet been widely adopted into routine clinical practice and will likely remain beyond the reach of most pathology laboratories for the foreseeable future. Automated assessment of Ki-67 is generally available with the most commonly used digital pathology systems [157], but the software is typically expensive and may be marketed only for research rather than clinical applications. The IKWG has proposed guidelines for setting up automated digital image analysis using the freely available Qu Path software platform [180], and there are some data to suggest that these guidelines are reproducible [107]. However, these guidelines involve the generation of new scoring algorithms using local data, rather than using existing algorithms that take advantage of the extensive research already performed to date. Furthermore, the guideline instructions may be difficult to follow for groups lacking individuals with substantial pathology informatics and/or image analysis expertise, and specifically caution against the routine use of the algorithms in high volume settings, such as what would be encountered in a busy pathology practice, without a high degree of understanding of the involved algorithms. Indeed, none of the automated analysis studies reviewed by the IKWG include detailed plans for inter-platform and inter-laboratory standardization. Therefore, substantial work needs to be completed to improve the ease in which pre-developed, well-tested automated digital image analysis algorithms can be implemented into pathology practices, presuming local algorithm validation but not local algorithm development, before the promise of this technology can be fully realized.

### 4.5. Potential Alternative/Complementary Assays to Ki-67 Immunohistochemistry

In addition to the extensive research that has been undertaken to standardize and improve the reliability and demonstrate the clinical utility of the Ki-67 immunohistochemical assay in breast cancer, there has also been much research undertaken to try to find alternative methods of measuring Ki-67 or identify new markers that can be used in conjunction with Ki-67. For instance, quantification of Ki-67 via RT-qPCR of *MKI67* mRNA expression has been proposed as alternative to either manual or automated measurement of Ki-67 immunohistochemistry [83,181,182,183], with some suggestion that Ki-67 RT-qPCR is better correlated than immunohistochemistry with clinical outcomes, such as pathologic complete response and distant disease-free survival [181,182]. Because of its inherently quantitative nature and lack of reliance on observer interpretation, RT-qPCR has been proposed to overcome the inherent limitations of Ki-67 immunohistochemistry. However, cost and lack of widespread availability remain significant barriers to implementation, and it is unclear whether cutoffs designed for immunohistochemistry-based assays would remain clinically meaningful when assessed using RT-qPCR.

In regard to potential alternative biomarkers, minichromosome maintenance 6 (MCM6) has recently been proposed as a marker with higher expression in more aggressive breast cancers—including luminal B, HER2-enriched, and triple-negative cancers—with discrimination performance similar to or perhaps better than Ki-67 [184,185]. Neurokinin 1 receptor (NK1R) has also been proposed as a potential prognostic marker, associated with high Ki-67, high tumor grade, and poor prognosis in breast cancer patients [186]. BCL-2 has been proposed as an additional prognostic marker that can be used in conjunction with ER, PR, HER2, and Ki-67 to predict the Oncotype DX breast cancer recurrence score [187]. Cyclin A and topoisomerase II alpha are additional proliferation markers that could be used in conjunction with Ki-67, particularly in cases with borderline Ki-67 values [188,189]. Some authors have identified microRNA signatures that are associated with Ki-67 levels and breast cancer prognosis [72,190,191]. Although an exhaustive discussion of all promising new biomarkers is beyond the scope of this review, given the above discussed challenges with effective utilization of Ki-67 in clinical practice, the potential to identify a marker or set of markers that can perform better than Ki-67 remains an active and important area of research.

## 5. Conclusions

Although the scientific community is closer than ever before in accepting Ki-67 as a prognostic and predictive biomarker in the care and treatment of breast cancer patients, several research issues remain open [2]. Improvements in precision achieved through the application of automated and digital scoring systems will help to further integrate these newer technologies into the pathology laboratory, leading to more time-efficient evaluation of Ki-67 and other breast cancer biomarkers. Similar to the confirmation of HER2 IHC results with in situ hybridization (ISH), correlation of clinical outcomes with Ki-67 as determined by gene expression may lead to more accurate definition of Ki-67 thresholds for prognosis and predictive therapy. The use of serial Ki-67 analyses to determine potential early treatment response endpoints for novel therapeutic agents is of research interest [2]; however, better determination of specific cutoff points related to particular clinical outcomes will be critical for this to occur, and for the future clinical utility of Ki-67. Most importantly, is the reproducibility of scoring expression associated with clinical outcomes, in particular when they are within the intermediate range of 10% to 25%, the range which has caused the most problems with inter-observer variability [40,192,193], and which accounts for a substantial fraction of clinical cases encountered in practice. In particular, a more simplified version of the IKWG scoring method that can be used in the routine pathology laboratory setting but is still accurate with the FDA-approved companion diagnostic assay at the threshold of ≥20% is greatly needed.

The future application of a Ki-67 threshold for clinical decision making in the management of high-risk ER-positive breast cancer in current pathology laboratories will depend on the standardization of pre-analytic, analytic, and post-analytic factors involved with Ki-67 testing. As was true for ER, PR, and HER2, pathology laboratories have the opportunity to make the necessary changes to help increase the reliability of Ki-67 as a biomarker with validated clinical utility, which will allow for its widespread adoption in clinical practice. Future studies using centrally confirmed standardized Ki-67 assays which test Ki-67 expression at multiple cutoff points, with assessment of outcomes, are critical in order for Ki-67 to gain international guideline consensus and acceptance as to its prognostic and predictive clinical utility in the care and treatment of breast cancer patients. 

## Figures and Tables

**Figure 1 cancers-15-00808-f001:**
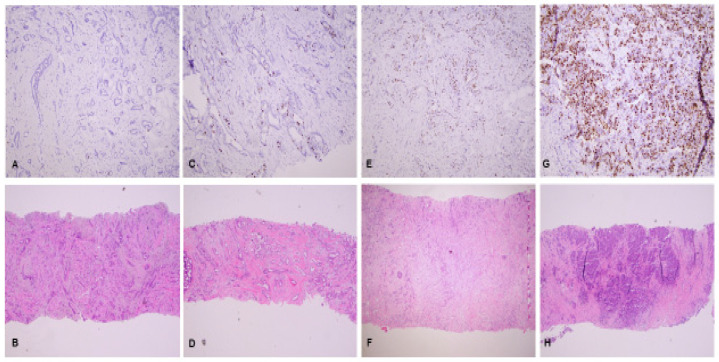
(**A**–**H**): Ki-67 staining (10×) category fields based on the International Ki-67 Working Group recommendations, with corresponding hematoxylin and eosin (H&E), (4×) staining. (**A**,**B**): Negative/Very Low (<1%) Ki-67 staining, with corresponding (H&E); (**C**,**D**): Low (≤5%) Ki-67 staining, with corresponding (H&E); (**E**,**F**): Medium (6–29%) Ki-67 staining, with corresponding H&E; (**G**,**H**): High (≥30%) Ki-67 staining, with corresponding (H&E).

**Figure 2 cancers-15-00808-f002:**
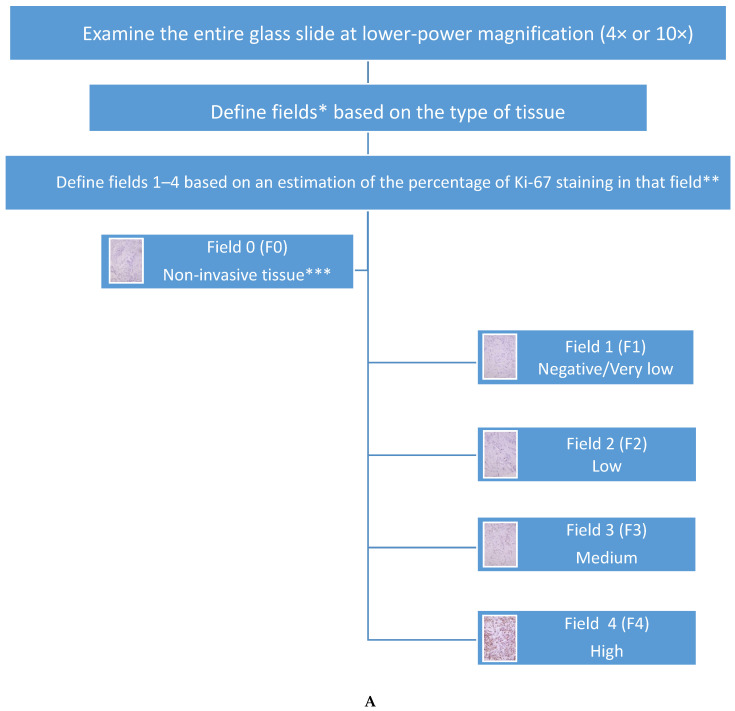
(**A**): Ki67-QC International Working Group whole section scoring protocol (global method): Specification of tissue fields and Ki-67 staining percentage in invasive tumor fields. * Fields are either non-invasive tissue (Field 0) or invasive carcinoma tissue (Fields 1–4). ** The percentage of Ki-67 staining in each invasive carcinoma field (fields 1–4) represents relative determinations of Ki-67 staining as a percentage of *all* the tissue on the glass slide (F0, F1, F2, F3, F4). Percentages should be considered based on the IKWG considerations of a low (≤5%) and high (≥30%) Ki-67 expression. Very low might be considered as <1%. *** Non-invasive tissue includes non-tumor tissue *and* in situ carcinoma. (**B**): Ki67-QC International Working Group whole section scoring protocol (global method): Calculation of overall Ki-67 staining percentage. * The percentage of each field (F0–F4) represents the area of that field relative to the area of the entire slide. For example, the percentage of invasive carcinoma field 1 (%F1) = %F1 = (^area of F1^/_area F0 + area F1 + area F2 + area F3 + area F4 + area F5_) × 100; ** For example, the relative percentage of invasive tumor nuclei in invasive carcinoma field 1 (%F1) = (^% F1^/_%F1 + %F2 + %F3 + %F4_) × 100; *** All invasive carcinoma fields may not be present (i.e., only fields of low and high Ki-67 staining may be present); however, in order to count a total of up to 400 cells, a total of four (4) invasive carcinoma fields must be considered. If a particular invasive carcinoma field is not present, then the count for that invasive carcinoma field would be allocated to another invasive carcinoma field that is present, per the IKWG Step 2 protocol: https://www.ki67inbreastcancerwg.org/wp-content/uploads/2018/12/Ki67-Phase-3b-WS-protocol-v1.pdf (accessed on 23 January 2023); **** In each field, count until either 100 invasive tumor nuclei have been counted, or all invasive tumor nuclei in the entire scoring field have been counted, whichever comes first. ***** The IKWG weighted Ki-67 score considers the effects of Ki-67 heterogeneity. IKWG unweighted Ki-67 = (^total # of positive Ki-67 cells from all fields^/_total # of invasive tumor cells from all fields_) × 100; IKWG weighted Ki-67 = ∑ _(%F1, %F2, %F3, %F4)_ %F (^# of positive Ki-67 cells in %F^/_# of invasive tumor cells in %F_) × 100.

**Figure 3 cancers-15-00808-f003:**
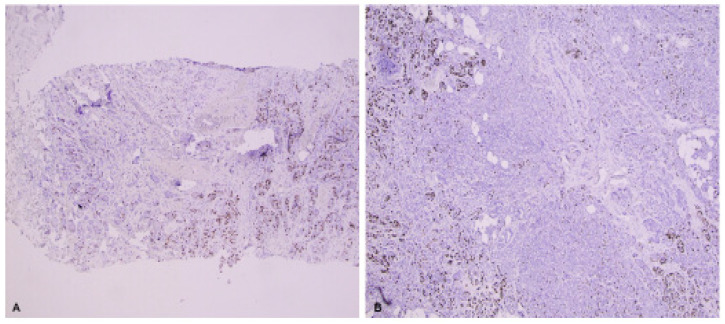
(**A**,**B**): Ki-67 immunohistochemical stain of breast cancer showing heterogeneous expression of Ki-67(10×). (**A**): Biopsy material; (**B**): Whole excision material.

**Table 1 cancers-15-00808-t001:** Summary of major meta-analyses of Ki-67 and breast cancer prognosis.

Meta- Analysis	Number of Studies	Range of Ki-67 Cutoffs	Range of *n*	Study Population(s)	Findings
Mirza et al. [44]	5	1–25%	212–674	Node-negative breast cancer	All studies associated with either DFS, OS, or both
de Azambuja et al. [45]	46	3.5–34%	47–863	Node-negative and -positive breast cancer	Reported overall HR = 1.93 (1.74–2.14) for DFS, and HR = 1.95 (1.70–2.24) for OS, in all patients
Stuart-Harris et al. [50]	43	0–28.6%	100–942	Node-negative and -positive breast cancer	Reported overall HR = 2.18 (1.92–2.47) for DFS, and HR = 2.09 (1.74–2.52) for OS, in all patients

**Table 2 cancers-15-00808-t002:** 2019 International Ki-67 in Breast Cancer Working Group Recommendations for the analytic validation of Ki-67 [2] *.

Studies should include a sufficiently large number of participating scorers to represent variability inherent in a broad cross-section of pathology interpretations.
Observers performing the scoring in test validation studies need to follow pre-specified training methods and score independently and, in a fashion, blinded to others’ scores.
A sufficient number of specimens should be included to have adequate statistical power, and the specimens should represent the entire dynamic range of the assay (in the case of Ki-67 IHC, 0–100%)
Although the expected implementation of tests is often categorical, based on one or more cutpoint(s), most tumor biomarkers (including Ki-67) are continuous variables, and data for assessing analytical validity should be captured as such. Doing so will allow for parametric tests that maximize information, and for results to be transposed to alternative cutpoints of clinical relevance. The data distribution for Ki-67 is log-normal, meaning that log transformation is required to satisfy the normal distribution and constant variance assumptions underlying common parametric statistical tests.
Studies using biospecimens or linking to prognosis should adhere to Biospecimen Reporting for Improved Study Quality (BRISQ) and Reporting Recommendations for Tumor Marker Prognostic Studies (REMARK) guidelines (including such important features as transparent and detailed reporting of scoring methods so others can apply the system exactly and pre-specified metrics of success, ideally with independent statistical analysis.

* Adapted from the 2019 International Ki-67 in Breast Cancer Working Group Recommendations [2].

**Table 3 cancers-15-00808-t003:** Factors to consider in the inter-laboratory reproducibility of Ki-67.

	Specific IKWG Recommendations	Comment
Pre-analytic Variables		
Core vs. Excision	Yes	Core biopsies are preferred. Serial analysis of Ki-67 should be performed on the same specimen type.
Pre-fixation Delays	Yes	Follow ASCO/CAP guidelines for breast tissue handling [134,135]. Ethanol-fixed or decalcified preparations should not be used.
Storage Time	No	Avoid prolonged exposure to air of cut sections on glass slides.
Analytic Variables		
Antigen Retrieval	Yes	High-temperature antigen retrieval should be mandatory.
Antibody Specificity	No	MIB1 is recognized as the most widely validated antibody.
Colorimetric Detection	No	Polymer detection on automated platforms is more sensitive than avidin-biotin systems.
Counterstain	Yes	All negative nuclei should be counterstained.
QA/QC Control	Yes	QA/QC control should be established and maintained in each laboratory and systematically maintained. Quantitative external quality assessment should be established, and participation should be mandatory.
Scoring Interpretation		
Method	Yes	Calculate the percentage of positive invasive carcinoma cells by counting all positive and negative invasive carcinoma cells within the examined region of the slide. Do not consider intensity.
Region of Slide	No	A global counting method appears to have higher reproducibility than a hot spot counting method.
Digital Imaging	No	Evidence to date suggests that automated scoring is not worse than standardized visual scoring for core-cut biopsies.
Data Capture	Yes	Ki-67 data should be captured as a continuous percentage variable, with log transformation for parametric statistical testing.

Adapted from the 2019 International Ki-67 in Breast Cancer Working Group Recommendations [2]. Abbreviations: ASCO = American Society of Clinical Oncology; CAP = College of American Pathologists; QA = Quality assurance; QC = Quality control.

## Data Availability

The data presented in this study is available within the article.

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
