# Peer review of "The Evolution of Ki-67 and Breast Carcinoma: Past Observations, Present Directions, and Future Considerations"

_cancers, 2023, doi:10.3390/cancers15030808_

Round 1
Reviewer 1 Report
The manuscript entitled “The Evolution of Ki-67 and Breast Carcinoma: Past Observations, Present Directions, and Future Considerations” described the past, present and future directions of Ki-67 as a diagnostic, prognostic and predictive biomarker for breast cancer. As a crucial and widely-used biomarker, Ki-67 has been strongly associated with cancer development. The topic is of great value for the readership; however, the manuscript itself is lack of organization and summaries, and it needs a lot of modifications before it can be considered for publication. Please see the detailed comments below:
1. The major issue of the manuscript is lack of classification and summary by the authors, whether by Ki-67 cut off points and range, by clinical application categories, breast cancer clinical stages, or breast cancer subtypes. It’s not well organized. For example, in the third section “Ki-67: present direction”, authors provided a detailed description of those mate-analysis studies from 2002, 2007 and 2008, and those clinical studies in the following paragraphs as well. It would be better if the authors can summarize their own points, give some examples future analysis and discuss the influence of those studies results, instead of simply going over those studies one by one and repeating the results or conclusions found in those studies. Subdivide into categories would make the manuscript better organized. A table to summarize all those studies would be helpful and please include the cuff off point of Ki67.
2. The description of the development of IKWG and its scoring protocols need be more concise and comprehensive as a review paper.
3. There were two review papers recently published in Ther Adv Med Oncol. entitled” Clinical validity and clinical utility of Ki67 in early breast cancer” (ref 41) and in Cancers (Basel) entitled “Ki-67 as a Prognostic Biomarker in Invasive Breast Cancer” Their papers are better organized and well presented. Please reference their publications and look deeper into Ki-67 application in different breast cancer subtypes, stages and treatment regimens, such as neoadjuvant chemotherapy, postoperative chemotherapy, endocrinotherapy, and targeted therapy.
4. Are there any biomarkers can be combined to indicate the clinical results with Ki-67 or even replace Ki-67 as a better prognostic and predictive biomarker? Please add to the discussion as well.
e.g. Sadeghian D, Saffar H, Mahdavi Sharif P, Soleimani V, Jahanbin B. MCM6 versus Ki-67 in diagnosis of luminal molecular subtypes of breast cancers. Diagn Pathol. 2022 Feb 6;17(1):24. doi: 10.1186/s13000-022-01209-4. PMID: 35125121; PMCID: PMC8818166.
e.e. Al-Keilani MS, Elstaty R, Alqudah MA. The Prognostic Potential of Neurokinin 1 Receptor in Breast Cancer and Its Relationship with Ki-67 Index. Int J Breast Cancer. 2022 Apr 4;2022:4987912. doi: 10.1155/2022/4987912. PMID: 35419208; PMCID: PMC9001113.
5. Figure 1 and 3: please add the scale bars
6. Page 2, line 3: many or countless?
Reviewer 2 Report
This review manuscript conducted by Turner et al. provided a historical overview and future directions on the use of Ki67 in breast cancer. The authors structured the manuscript to Past Observations, Present directions, and Future Considerations. The topic is very current and compelling. Overall, the work is well organized, as it highlights the key points of promises and challenges implementing Ki67 for the daily practice of breast cancer diagnostics. Furthermore, the paper provides the summary of the scoring method by International Ki67 in Breast Cancer (IKWG). The manuscript is easy to follow even if the professional is less involved in this field. However, I believe that authors should put more focus on the future directions. I have major comments enumerated below:
1) The authors should focus on the potential issues for using Ki67 as companion diagnostic test approved by the FDA. Following the lines of argument now laboratories would have to launch the DAKO assay, even if laboratories have implemented a valid assay using another platform. Please elaborate on this in the manuscript.
2) Regarding the monarchE trial and setting the 20% cutoffs. In Monarch E the Hazard ratio for IDFS in the ITT population was 0.713 (95%CI 0.583-0.871) whilst in the Ki67>20% group it was 0.691 (95% CI 0.519-0.920) – whilst the Ki67 low group was not presented –one can argue how the HR for the Ki67 low group could possibly support a significant treatment and marker interaction given the low difference in HR (0.022) between Ki67 high vs all cases. In Penelope B (a negative study) the HR for Ki67 LOW was 0.873 (0.654-1.16) and for Ki67 HIGH was 1.02 (0.718-1.46) again NO evidence of a treatment by marker interaction and if anything, it would suggest benefit in Ki67 LOW cases. Based on this above, this is the first required companion diagnostic test with only a prognostic, but not a predictive role. Is the most relevant clinical threshold really the 20% cutoff?
I recommend the authors to write about these arguments, especially, because in Europe, it is not a requirement to administer abemaciclib based on a Ki67 test.
3) Regarding the analytic and clinical validity of the Ki-67 results from the monarchE clinical trial. The SSED report on reproducibility depends on results from 3 trained pathologists scoring 60 pre-stained sections 3 times. Statistically, this study design is insufficient to determine the level of variability likely to occur within- or between-pathologists in real world applications. There are studies demonstrating with more than 8 pathologists for other biomarkers have shown dramatically poorer agreement than studies with only 2 or 3 pathologists (PMID: 34897773; PMID: 32300181; PMID: 35816627).
Furthermore, the Polewski data showed good concordance both within- and between-pathologists, however one pathologist scored approx.7-8% lower on one run than on the other 2 runs, such that a Ki67 value of 20% in 2 sets would have read at approx.12-13% in the other (Polewski et al supplemental Figure 6)
Based on these arguments, I recommend to further elaborate on the sections page 14.
4) I think the paper would be enhanced by mentioning the possibilities and difficulties of digital Ki67 scoring for pathology departments. There are several studies available comparing pathologist reads against machine scoring and there are studies now comparing different AI platforms. However, little focus has been put so far in the literature on how to implement and validate these tools. For this reason, could you please elaborate on the analytical and clinical aspects on digital scoring in Ki67?
5) On page 12, regarding the higher variability in Ki67 scoring between 6-29%. It is important to mention that the IKWG is not claiming there is lower variability in Ki67 scores when they are below 5% or above 30%, but rather that those values are sufficiently different from prognostic decision cutoffs that, despite the residual analytical variability that remains after standardization, results are sufficient for the purpose of deciding on the need for adjuvant chemotherapy in anatomically favorable ER-positive and HER2-negative patients (PMID: 34003287).
Round 2
Reviewer 1 Report
Thank you for all the modifications the authors made and I am satisfied with the current revision.
Reviewer 2 Report
Thank you for your response! The authors have addressed all my questions and comments.